# At freedom's edge: Belief in free will during the COVID-19 pandemic

**Elizabeth Seto** [ORCID] *

Department of Psychology, Colby College, Waterville, Maine, United States of America

* seto.elizabeth@gmail.com

## Abstract

Among life-and-death health concerns surrounding the COVID-19 pandemic were frustrations about the loss of personal freedom due to emergency quarantine. To test these perceptions, two studies examined whether belief in free will was resilient during different points of the pandemic. In Study 1, conducted in 2020, participants completed a writing task describing their lives before the COVID-19 pandemic, during the COVID-19 pandemic while under emergency quarantine, and during the COVID-19 pandemic while under state re-openings. Following each task, they completed belief in free will measures. Results indicated that free will beliefs were higher before the pandemic than during emergency quarantine. Free will beliefs were also greater during state re-opening than during emergency quarantine. Belief in free will did not differ between pre-pandemic and state re-opening. Study 2 replicated and extended these effects two years later. These findings highlight the brief loss of freedom during COVID-19 as well as the resiliency of agentic control.

## Introduction

On March 11, 2020, the World Health Organization declared COVID-19 a pandemic. The dangerous spread of the virus and the exponential number of deaths prompted countries across the world to engage in lockdowns. This included stay-at-home mandates, closing of businesses, working from home, and maintaining physical distance from others [1]. While life-and-death health concerns were at the forefront of the pandemic, frustrations about the loss of personal freedom were also prevalent. Most notably, there was forceful opposition towards emergency quarantine [2], and fears that freedom would continue to decline as the pandemic waned on [3]. The current research tested the reality of these perceptions by examining whether belief in free will was resilient during different points of the COVID-19 pandemic.

### Belief in free will

The ability to make your own choices and determine your own outcomes underlie the basic notion of belief in free will [4]. Most people believe in free will [5], and intuitions about free will have been demonstrated cross-culturally [6]. Free will beliefs are widespread largely because they invoke feelings of personal agency [4]. That is, people highly value control of

**Data Availability Statement:** Pre-registration, study materials, data sets, and supplementary analyses for both studies can be found on OSF: https://osf.io/h8935/.

**Funding:** The author disclosed receipt of the following financial support for the research,

authorship, and/or publication of this article: This work was supported by grant funding from the Colby College Social Science Division. The funders had no role in study design, data collection and analysis, decision to publish, or preparation of the manuscript.

**Competing interests:** The authors have declared that no competing interests exist.

their own lives. Research has found that belief in free will is positively associated with agentic traits such as self-efficacy [7] and internal locus of control [8]. Moreover, inducing disbelief in free will reduces components of self-agency [9].

Varying belief in free will has other meaningful consequences. For instance, encouraging disbelief in free will increases aggression [10] and promotes conformity [11]. Attenuating belief in free will diminishes true self-knowledge [12], feelings of gratitude [13], and the pursuit of meaningful goals [7]. The costly outcomes of altering belief in free will suggests that exercising agentic control over a situation is important to most people. While research has established the importance of believing in free will, research has yet to explore how resilient these beliefs are. The present research investigated how belief in free will changed in the real world under the unique context of the COVID-19 pandemic.

## Restrictions during the COVID-19 pandemic

The COVID-19 pandemic was an unprecedented life event that brought the world to a sudden halt. In an effort to prevent virus transmission and protect public health, policymakers around the world enacted freedom-restricting measures [14]. Lockdowns and travel restrictions limited mobility. Mask mandates and social distancing requirements inhibited social interaction. Prevention of mass gatherings hindered community building. While research has shown that the trade-off between freedom and security is effective in thwarting the spread of viruses [15], the perceived loss of individual freedom during this historical time is still contentious.

Feelings of restricted freedom was experienced on a global level during the pandemic. People in U.K. reported frustration over the lack of agency in maintaining social connections during initial phases of the lockdown [16]. Similarly, employees in the U.S. experienced a threat to their autonomy during the early stages of the pandemic when the government issued stay-at-home and social distancing guidelines [17]. During the first wave of the COVID-19 pandemic, there were even fears about permanent restrictions on freedom after experiencing strict lockdowns in Spain, the U.K., and Italy [3]. While infection from the virus was a top concern for many, constraints on freedom and an uncertain future also remained people's minds.

## Belief in free will during the COVID-19 pandemic

Discussions about restrictions on freedom call into question whether belief in free will diminished during the COVID-19 pandemic and whether it ever restored back to pre-pandemic levels. Although belief in free will is considered a stable construct, research has shown that free will beliefs also shift under different circumstances [4]. Reports of reduced agency and autonomy during the pandemic [16, 17] also reflect attributes closely aligned with belief in free will [18], suggesting that free will beliefs were likely to change during this pivotal time in history.

We predicted that free will beliefs would be altered during the COVID-19 pandemic. Specifically, belief in free will would be highest pre-pandemic before COVID-19 restrictions became a notable constraint on people's lives. During the pandemic, belief in free will would take a downward trajectory due to freedom-restricting measures such as quarantines, lockdowns, and social distancing requirements [14]. Finally, belief in free will would revert closely back to pre-pandemic levels when public health measures eased, and people adapted to living life with the threat of the virus contained.

## Overview of the current studies

In the present research, we examined whether belief in free will changed across different time points of the COVID-19 pandemic. Study 1, conducted in 2020, assessed free will beliefs before the pandemic, during emergency quarantine, and under state re-openings. Study 2, conducted

in 2022, measured free will beliefs before the pandemic, during emergency quarantine, under state re-openings, and two years into the pandemic. Across both studies, we predicted a decline in belief in free will during the early pandemic with emergency quarantine in place and then a restoration of belief in free will after state re-openings. It is important to note that we utilized a retrospective approach across both studies to assess belief in free will. As participants were from the United States, and the implementation and cessation of COVID-19 lockdowns occurred on different state timelines, retrospective reporting was the best option for data collection at the time. Pre-registration, study materials, data sets, and supplementary analyses for both studies can be found on OSF.

## Sample size determination

In Studies 1 and 2, an a priori power analysis was conducted to achieve a power equal to .95 with a with a small effect size (partial $\eta^2$ = .02) at an alpha level of .05. A sample size of 128 was required in Study 1. We collected more than 1.5 times the sample size requirement to maximize power and in anticipation of attrition. A sample size of 107 was required in Study 2. We collected more than 2 times the sample size requirement to maximize power and in anticipation of attrition. Data collection was terminated once these goals were met. Following our pre-registered data exclusion procedures, duplicate IP addresses, same geolocations, incomplete study responses, and incomprehensible study responses (e.g., bots) were excluded from final data analysis.

## Ethics statement

The Colby College IRB reviewed and approved this body of research. The IRB approval number is 2020–027. Informed consent was obtained in electronic written form, and data was analyzed anonymously.

## Study 1

Study 1 investigated whether different stages of the COVID-19 pandemic influenced belief in free will. Data was collected in 2020, the same year COVID-19 was declared a pandemic by the World Health Organization. We predicted that belief in free will would be highest before the pandemic, followed by state re-opening, and during emergency quarantine. In other words, we expected a decline and progression of belief in free will during the initial stages of the COVID-19 pandemic.

### Method

**Participants.** Two-hundred and four participants (125 female, 77 male, 2 non-binary; $M_{age}$ = 35.50, $SD_{age}$ = 12.68, range 18–76; 138 White, 30 Black or African-American, 19 Asian or Asian-American, 9 more than one race, 3 Hispanic, Latinx, or Mexican American, 3 Indian, 2 American-Indian/Alaskan Native) from the United States were recruited from Amazon Mechanical Turk (MTurk) using CloudResearch from October 14, 2020 –November 9, 2020 and compensated with $1.25. We recruited CloudResearch-Approved Participants to ensure high data quality [19]. Additional demographic variables including religiosity, political orientation, social class background, and subjective socioeconomic status can be found in our supplementary materials.

**Materials and procedure.** Participants provided their written consent online, completed the measures described in order below, and were debriefed.

*Pandemic time manipulation.* Participants were instructed to think about themselves at three different time points during the COVID-19 pandemic: normal life ("before the COVID-19 pandemic while under normal circumstances"), quarantine life ("during the beginning of the COVID-19 pandemic while under any state of emergency quarantine"), and re-opening life ("during the ongoing COVID-19 pandemic while under any state re-opening"). They were told to consider their interactions with family, friends, and co-workers, hobbies they engaged in, and any activities they pursued during each time period. Then, participants wrote a brief description of their lives during each time period of the COVID-19 pandemic.

*Belief in free will.* Following each COVID-19 life description, participants completed the 7-item belief in free will subscale of the Free Will and Determinism Plus Scale ("People have complete free will."; FAD+) [8] to assess belief in free will across the three different time points of the COVID-19 pandemic. The subscale used a 7-point scale (1 = *strongly disagree*, 7 = *strongly agree*), and items were averaged to produce a composite score where higher values indicated greater belief in free will.

## Results

A repeated-measures ANOVA was conducted to determine if there was a difference in belief in free will across the different time points of the COVID-19 pandemic. There was a significant difference in belief in free will across time points ($F(2, 406) = 8.81$ $p < .001$, partial $\eta^2 = .04$). Pairwise comparisons with Bonferroni corrections indicated that participants reported greater belief in free will before the COVID-19 pandemic ($M = 4.95$, $SD = 1.04$, $p = .041$) and after the state re-opening ($M = 5.00$, $SD = 1.10$, $p < .001$) than during emergency quarantine ($M = 4.86$, $SD = 1.15$). Belief in free will did not differ before the COVID-19 pandemic and after state re-opening ($p = .384$). See Fig 1.

## Study 2

Study 1 found that belief in free will changed during the initial stages of the COVID-19 pandemic. In line with our predictions, free will beliefs were higher pre-pandemic than during

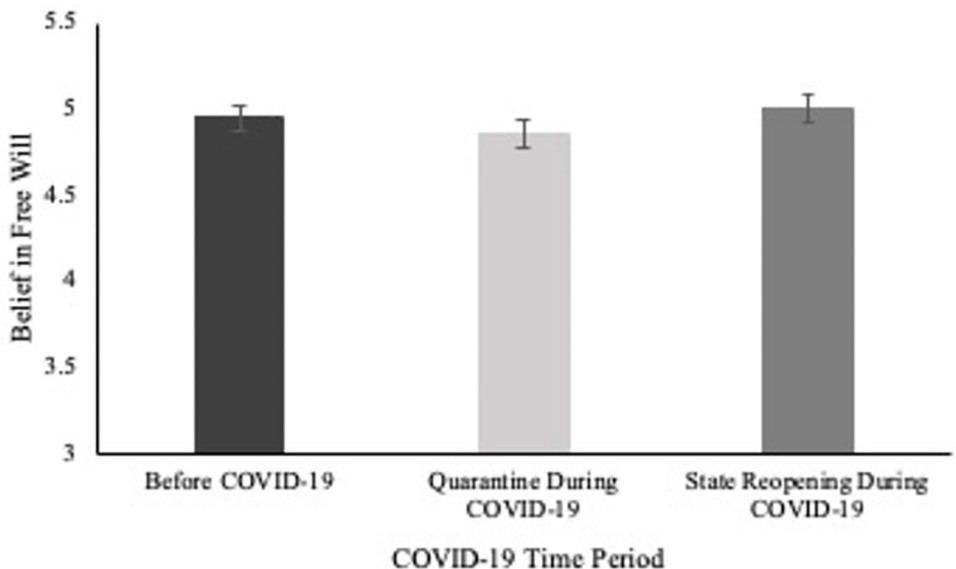

**Fig 1. Differences in belief in free will as a function of COVID-19 pandemic time points in Study 1.** Standard errors are represented by the error bars.

emergency quarantine. Belief in free will was also greater during state re-opening than during emergency quarantine. Surprisingly, free will beliefs did not differ between pre-pandemic and state re-opening. While belief in free will declined temporarily during emergency quarantine, it was restored to pre-pandemic levels after state re-opening.

To instill confidence in our findings, Study 2 attempted to replicate and extend these effects two years into the pandemic. Data was collected in 2022, two years after COVID-19 was declared a pandemic by the World Health Organization. We predicted that belief in free will would be lower during quarantine compared to life pre-pandemic, during state re-opening, and two years into the pandemic. We also predicted that belief in free will would be similar across the pre-pandemic time period, state re-opening, and two years into the pandemic. In other words, we expected a decline, progression, and plateau of belief in free will.

## Method

**Participants.**   Two hundred and twenty participants (151 female, 68 male, and 1 non-binary; $M_{\text{age}}$ = 38.82, $SD_{age}$ = 12.67, range 19–74; 178 White, 25 Black or African-American, 18 Latino/Latina, Hispanic, or Latin American, 12 Asian or Asian-American, 2 Native-American, American-Indian, or Alaskan Native, and 1 other) from the United States were recruited from MTurk using CloudResearch from August 26, 2022 –August 27, 2022. CloudResearch-Approved Participants completed the study for $2.50. Additional demographic variables including religiosity, political orientation, social class background, and subjective socioeconomic status can be found in our supplementary materials.

**Materials and procedure.**   Participants provided their written consent online, completed the measures described in order below, and were debriefed.

*Pandemic time manipulation.* Participants were instructed to think about themselves at four different time points during the COVID-19 pandemic. The first three time points (normal life, quarantine life, and re-opening life) were identical to Study 1. The fourth time point was new normal life ("during the ongoing COVID-19 pandemic two years later"). As in Study 1, participants wrote a brief description of their lives during each time period of the COVID-19 pandemic.

*Belief in free will.* Participants completed the same belief in free will measure for each COVID-19 pandemic time period from Study 1.

## Results

A repeated-measures ANOVA was conducted to determine if there was a difference in belief in free will across the different time points of the COVID-19 pandemic. There was a significant difference in belief in free will across time points ($F(3, 559)$ = 13.23, $p < .001$, partial $\eta^2$ = .06; sphericity not assumed). Pairwise comparisons with Bonferroni corrections indicated that participants reported greater belief in free will before the COVID-19 pandemic ($M$ = 5.05, $SD$ = .89, $p$ = .045) and after the state re-opening ($M$ = 5.12, $SD$ = .97, $p < .001$) than during emergency quarantine ($M$ = 4.94, $SD$ = 1.00), replicating the findings in Study 1. Participants also reported greater belief in free will two years later into the COVID-19 pandemic than during emergency quarantine ($M$ = 5.15, $SD$ = .96, $p < .001$). Replicating Study 1 again, belief in free will did not differ before the COVID-19 pandemic and after state re-opening ($p$ = .389). Belief in free will did not differ after state re-opening and two years later ($p$ = 1.00). Participants reported greater belief in free will two years later into the COVID-19 pandemic than before the COVID-19 pandemic ($p$ = .046). See Fig 2.

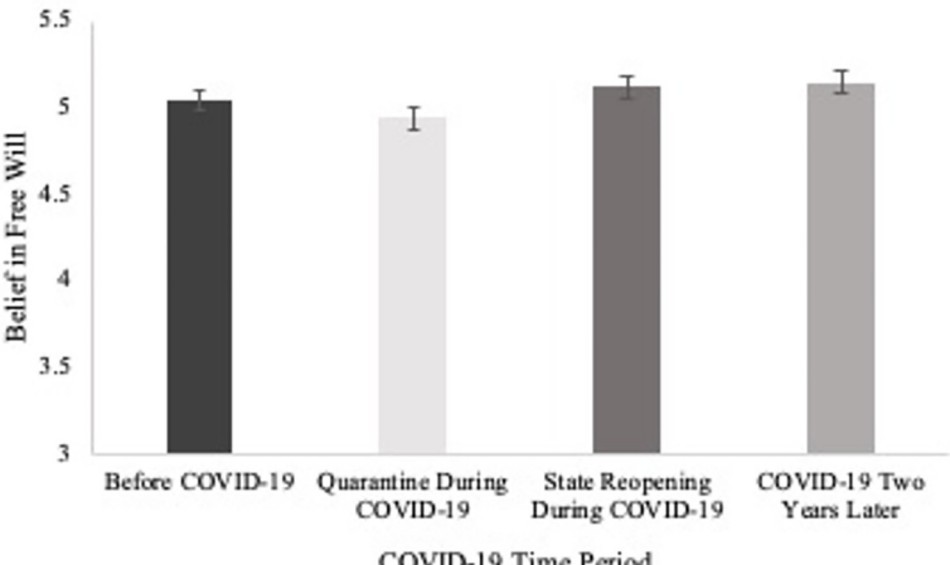

**Fig 2. Differences in belief in free will as a function of COVID-19 pandemic time points in Study 2.** Standard errors are represented by the error bars.

## Discussion

The present research examined whether belief in free will was resilient during the COVID-19 pandemic. In Study 1, conducted in 2020, belief in free will declined temporarily during emergency quarantine, but was restored to pre-pandemic levels after state re-opening. Study 2, conducted in 2022, replicated these effects and also found that belief in free will remained stable after state re-opening and two years after COVID-19 was declared a pandemic. Although fears about the loss of freedom during the COVID-19 pandemic were warranted, people's belief in free will were resilient against an unprecedented public health crisis.

Integral to the belief in free will is the view of an agentic self, unconstrained by environmental influences and having the ability to make choices [4]. Since most people report believing in free will [5], we expected that free will beliefs would shift across different time points of the COVID-19 pandemic. Both studies, conducted two years apart, generally supported this prediction. During pre-pandemic, before COVID-19 restrictions were implemented, most people experienced personal agency. Public health measures during the pandemic in the form of quarantine and lockdowns reduced agentic control and made people feel less certain about how they could determine their own future. State re-openings seemed to restore belief in free will likely because security measures eased and a sense of choice regarding mobility and social interaction was imparted back to the public.

Both studies demonstrated the resiliency of free will beliefs. In Study 1, we predicted that belief in free will would be highest before the pandemic, followed by state re-opening, and during emergency quarantine. Free will beliefs did not differ between pre-pandemic and state re-opening. Belief in free will also recovered and plateaued from state re-opening to two years into the COVID-19 pandemic in Study 2. These findings suggest that free will beliefs restore quickly and are generally stable unless subjected to circumstances out of personal control. Moreover, Study 2 found that belief in free will was higher two years later into the COVID-19 pandemic than before the COVID-19 pandemic. While we predicted that belief in free will would be similar across pre-pandemic, state re-opening, and two years into the pandemic,

psychological reactance might explain this finding. Research has found that when people experience restrictions on freedom, they behave in ways to regain their freedom [20]. It is possible that perceived freedom-restricting measures during quarantine and lockdown prompted people to reassert their personal agency more than they did during their pre-pandemic lives.

The resiliency of belief in free will is in line with research examining whether core values change across time. Core values tend to be highly stable, but also change in adulthood as a function of biological and physical maturation [21, 22]. Research has shown that the basic notion of free will develops between four to six years old [23, 24]. Additionally, in research involving experimental manipulation of free will beliefs using adult samples, belief in free will and control conditions tend to yield similar statistical findings (e.g., [10, 11]), suggesting that most people have baseline free will beliefs. These findings demonstrate that belief in free will is quite stable, but can change temporarily depending social context.

Our findings are also especially notable given that responses to COVID-19 lockdowns and restrictions tend to intersect with political values. While most people believe in free will [5], research has shown that free will beliefs are associated with greater conservative worldviews [25], and political differences in belief in free will stem from differences in how people assign moral issues [26]. Across both studies, participants were fairly politically diverse, and they reported being slightly liberal/moderate on average. This suggests that changes in free will beliefs during the COVID-19 pandemic are not necessarily associated with a specific political orientation. The reduction and restoration of belief in free will during the COVID-19 pandemic was a collective experience in the United States. This is perhaps unsurprising given that the United States is a nation that highly values freedom and individual rights [27]. The unexpected consequences of the COVID-19 pandemic made the loss of belief in free will more palpable.

While our findings suggest that belief in free will can recover from an unpredictable public health crisis, there are several limitations to our research. The current work is based on retrospective reports potentially subject to memory errors. While this is unlikely, given the timing of data collection for Study 1 and the replication and extension nature of Study 2, using experience sampling methodology could have alleviated concerns about memory accuracy. Participants were also from the United States and predominantly white. Future research might replicate these findings in nations with different cultural values and recruit more diverse social identities to enhance the generalizability of our findings. In a similar vein, countries across the world had varying responses to the COVID-19 pandemic including differences in levels of lockdown and disparities in lockdown implementation. It may be fruitful to explore whether belief in free will changed as a function of different lockdown policies. Finally, although the COVID-19 pandemic provided a unique historical context to examine the resiliency of belief in free will, future research could examine how free will beliefs change across other circumstances.

## Conclusion

The brief loss of freedom during the COVID-19 pandemic reminds us of how much we value agency in our lives. While belief in free will collectively plunged during the initial enactment of public health measures, it also quickly recovered and became more stable with time. This suggests that people truly believe and want to be the author of their lives. This sentiment is important to remember as we realize the consequences of this historical public health crisis in years to come.

## Author Contributions

**Conceptualization:** Elizabeth Seto.

**Data curation:** Elizabeth Seto.

**Formal analysis:** Elizabeth Seto.

**Funding acquisition:** Elizabeth Seto.

**Investigation:** Elizabeth Seto.

**Methodology:** Elizabeth Seto.

**Project administration:** Elizabeth Seto.

**Resources:** Elizabeth Seto.

**Software:** Elizabeth Seto.

**Supervision:** Elizabeth Seto.

**Validation:** Elizabeth Seto.

**Visualization:** Elizabeth Seto.

**Writing – original draft:** Elizabeth Seto.

**Writing – review & editing:** Elizabeth Seto.

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
