## [Decision Letter · Decision Letter 0]

25 Jan 2024

PONE-D-23-31527At freedom’s edge: Belief in free will during the COVID-19 pandemicPLOS ONE

Dear Dr. Seto,

Thank you for submitting your manuscript to PLOS ONE. After careful consideration, we feel that it has merit but does not fully meet PLOS ONE’s publication criteria as it currently stands. Therefore, we invite you to submit a revised version of the manuscript that addresses the points raised during the review process.

We look forward to receiving your revised manuscript.

Kind regards,

Ali B. Mahmoud, Ph.D.

Academic Editor

PLOS ONE

Journal Requirements:

3. Thank you for stating the following financial disclosure: "The author ES disclosed receipt of the following financial support for the research, authorship, and/or publication of this article: This work was supported by grant funding from the Colby College Social Science Division. "

Reviewers' comments:

Reviewer's Responses to Questions

**Comments to the Author**

1. Is the manuscript technically sound, and do the data support the conclusions?

Reviewer #1: Yes

Reviewer #2: Partly

2. Has the statistical analysis been performed appropriately and rigorously? 

Reviewer #1: Yes

Reviewer #2: Yes

3. Have the authors made all data underlying the findings in their manuscript fully available?

Reviewer #1: Yes

Reviewer #2: Yes

4. Is the manuscript presented in an intelligible fashion and written in standard English?

Reviewer #1: Yes

Reviewer #2: No

5. Review Comments to the Author

Reviewer #1: Dear Dr. Seto,

Thank you for the opportunity to review your work. I have minor comments regarding the paper, although before presenting them I just want to clarify one thing. I believe that resilience was the keyword that made this paper find me, so I am not an expert in free will. In this sense, my comments are about the use of resilience in the paper and some methodological aspects of the research.

1) Despite presenting data clearly, I missed if you collected any demographic data. Also, political and religious beliefs seem to be important when investigating free will (https://doi.org/10.1111/j.1467-6494.2012.00799.x). Did MTurk provide you with a sample with those different profiles? Also, in a country where COVID-19 and its restrictions were such a political topic, do you think political values help explain further the trends you have observed? In your view/interpretation of the data, democrats, republicans, and independents would have this stable trend because of American culture about free will? I think your discussion could benefit if you present your views on those aspects to the readers.

2) You indicate that your works aim to verify how resilient the free will attitudes are before and after pandemic. Although, in the discussion you say “fluctuated”. Now, since this is my turf, I would recommend you sticking to resilience for 2 reasons. First, when discussing resilience, we are considering how adaptative a behavior/cognition/emotion is when facing adversity. COVID-19 restrictions were an adversity in several levels and showing, as your data did it, that in similar contexts the beliefs levels decreased and increased according to context. So, it is not a fluctuation, it is a resilient response according to context. Further, core values seem to be more resilient in the sense that they do not show major changes during adulthood (https://journals.plos.org/plosone/article?id=10.1371/journal.pone.0289487 and https://doi.org/10.1177/0146167216639245 are 2 studies that show this trend). In this sense, your sudden change and return to baseline are interesting because it shows how fast-paced beliefs might change considering context, which is exactly how we are theorizing (and testing) the effects of resilience on cognition (attitudes, beliefs, etc). I missed a more in-depth discussion about those issues.

Sincerely,

Dr. Sidnei Priolo Filho

Reviewer #2: P10 The background section outlining freewill was good.

P10 ln 66 “what many perceived to be freedom-restricting measures” I fail to see how the authors can describe the measures as “perceived” freedom restricting. They were freedom restricting. Moreover it is difficult to accept the approach to the rest of the paper if the authors only feel they are testing a perception rather than an actual. Surely a strength of this work is testing actual not perception.

P11 ln 70 The statement that the evidence of lockdown is pivotal in defeating the virus is unconvincing.

The authors perhaps take a little too much time reiterating their hypothesis assumptions.

P12 a potential major flaw in the work is the authors present the studies as if in real time. The problem is unless the pre period work already existed phase one relies on asking people retrospectively how they felt before the pandemic, so is already subject to bias. Granted retrospective may have been the only option but they need to be more open and honest and explicit that their approach is retrospective.

P13 the participant details are too sparse to draw any conclusion about whether this was an unbiased and balanced sample. I’m not familiar with MTurk but were the participants from a certain social, ethnic, demographic, background etc. Cloud research itself introduces bias.

The most important finding is that both studies demonstrated a return to “belief in free will” at pre pandemic levels. Ie freewill belief has not diminished.

I feel the middle period in both studies “the pandemic” was considered in too trivial a fashion. The authors talk about relinquishing in freewill because of overriding proven health reasons. As very little was and still is proven in this period this is a shaky assumption at best. Further they totally fail to address the different methods of coercion adopted by governments and bodies such as the WHO. The UK used subvertive advertising and a government nudge department. NHSE have had to subsequently admit that claims unvaccinated people spread covid more than vaccinated were untrue.

The authors were very honest and clear about the limitations of their study.

It would have been good if they could also have addressed that governments because of such findings of - no lasting damage to freewill - could use such unevidenced measures of lockdown again especially given their conclusion.

The 2 figures illustrate their findings well and is powerful.

The paper is clear and simple. Perhaps over simplification is their enemy.

6. PLOS authors have the option to publish the peer review history of their article (what does this mean?). If published, this will include your full peer review and any attached files.

Reviewer #1: **Yes: **Sidnei Rinaldo Priolo Filho

Reviewer #2: **Yes: **Prof Marilyn James

---

## [Author Response · Author response to Decision Letter 0]

16 Mar 2024

Journal Requirements:

To the best of my knowledge, the manuscript meets PLOS ONE’s style requirements.

The raw data for this manuscript is available on Open Science Framework (OSF): https://osf.io/h8935/?view_only=981761395729438baa60e239df3cfd95.

3. Thank you for stating the following financial disclosure: "The author ES disclosed receipt of the following financial support for the research, authorship, and/or publication of this article: This work was supported by grant funding from the Colby College Social Science Division. "

The amended Role of Funder statement is included in the cover letter.

The full ethics statement is now included the Method section of both studies in the manuscript (p. 6-7 and 9).

Reviewer #1 Comments

1) Despite presenting data clearly, I missed if you collected any demographic data. Also, political and religious beliefs seem to be important when investigating free will (https://doi.org/10.1111/j.1467-6494.2012.00799.x). Did Murk provide you with a sample with those different profiles? Also, in a country where COVID-19 and its restrictions were such a political topic, do you think political values help explain further the trends you have observed? In your view/interpretation of the data, democrats, republicans, and independents would have this stable trend because of American culture about free will? I think your discussion could benefit if you present your views on those aspects to the readers.

In our original manuscript, we reported standard demographic information in both studies including gender, age, and race. We now report all identifications for race in the revised manuscript (p. 6 and 9), and descriptive statistics for demographic data including political orientation, religiosity, social class background, and subjective socioeconomic status can be found in a table in our supplementary materials on OSF.

We collected demographic information about religiosity (1 item; 1 = not religious at all, 7 = very religious) and political orientation (1 item; 1 = very liberal, 7 = very conservative). Across both studies, participants indicated being slightly less religious/neither religious or unreligious (M = 3.35, SD = 2.22, for Study 1; M = 3.47, SD = 2.12, for Study 2) and slightly liberal/moderate (M = 3.34, SD = 1.75, for Study 1; M = 3.58, SD = 1.80, for Study 2).

Carey and Paulhus (2013) found that belief in free will was positively correlated with religiosity and global conservatism, respectively. We conducted an ANCOVA controlling for both measures. In short, we found that the effect of time on belief in free will remained significant when controlling for religiosity (p = .001, for Study 1; p < .001, for Study 2). The effect of time on belief in free will controlling for political orientation was trending towards significance in Study 1 (p = .190) and reduced to marginal significance in Study 2 (p = .061). It is important to note that our 1-item measure of political orientation is likely different than the Carey and Paulhus’ (2013) measure of global conservatism. Unfortunately, the article does not indicate the specific items used in their research; therefore, it is hard to draw conclusions about the role of conservatism in our findings. Nevertheless, these results are now reported in our supplementary materials on OSF.

The proposition that political values could help explain the changes in belief in free will is an interesting possibility. Although most people believe in free will (Nahmias et al., 2005), and belief in free will is positively correlated with conservatism (Carey & Paulhus, 2013), recent research has found that political differences in belief in free will stem from differences in how people assign moral judgment (Everett et al., 2021). Participants across both studies were fairly politically diverse. They reported being slightly liberal/moderate on average. This suggests that people generally share a similar perspective that belief in free will shifted across the COVID-19 pandemic even across political divides. We agree that this is likely due to the importance the United States places on freedom. We now discuss this perspective in the Discussion (p. 12). 

2) You indicate that your works aim to verify how resilient the free will attitudes are before and after pandemic. Although, in the discussion you say “fluctuated”. Now, since this is my turf, I would recommend you sticking to resilience for 2 reasons. First, when discussing resilience, we are considering how adaptative a behavior/cognition/emotion is when facing adversity. COVID-19 restrictions were an adversity in several levels and showing, as your data did it, that in similar contexts the beliefs levels decreased and increased according to context. So, it is not a fluctuation, it is a resilient response according to context. Further, core values seem to be more resilient in the sense that they do not show major changes during adulthood (https://journals.plos.org/plosone/article?id=10.1371/journal.pone.0289487 and https://doi.org/10.1177/0146167216639245 are 2 studies that show this trend). In this sense, your sudden change and return to baseline are interesting because it shows how fast-paced beliefs might change considering context, which is exactly how we are theorizing (and testing) the effects of resilience on cognition (attitudes, beliefs, etc.). I missed a more in-depth discussion about those issues.

We thank the reviewer for bringing up this important distinction between resilience and fluctuation. We removed language regarding fluctuations in belief in free will throughout the manuscript. Additionally, we include greater discussion about the resiliency of belief in free will in the Discussion, citing these particularly helpful references on the stability of core values across the lifespan (p. 12).

Reviewer #2 Comments

P10 The background section outlining freewill was good.

Thank you.

P10 ln 66 “what many perceived to be freedom-restricting measures” I fail to see how the authors can describe the measures as “perceived” freedom restricting. They were freedom restricting. Moreover it is difficult to accept the approach to the rest of the paper if the authors only feel they are testing a perception rather than an actual. Surely a strength of this work is testing actual not perception.

We agree that the COVID-19 pandemic lockdown and measures reflected actual rather than perceived freedom-restricting measures. Therefore, the language “what many perceived to be” was deleted from this sentence (p. 3) and amended throughout the manuscript.

P11 ln 70 The statement that the evidence of lockdown is pivotal in defeating the virus is unconvincing.

In the Kucharski et al. (2020) article, the authors describe the effectiveness of isolation, testing, contacting tracing, and physical distancing to reduce the transmission of SARS-CoV-2. This approach was similarly undertaken by countries around the world during the COVID-19 pandemic. We amended the language from “pivotal” to “effective” to better capture these responses to the pandemic (p. 4).

The authors perhaps take a little too much time reiterating their hypothesis assumptions.

We believe it is important to highlight the reasoning behind our hypothesis that belief in free will would shift in response to the context of the COVID-19 pandemic. We decided to err on the side of clarity and have kept the current description of our hypotheses. 

P12 a potential major flaw in the work is the authors present the studies as if in real time. The problem is unless the pre period work already existed phase one relies on asking people retrospectively how they felt before the pandemic, so is already subject to bias. Granted retrospective may have been the only option but they need to be more open and honest and explicit that their approach is retrospective.

To increase transparency about our research approach, we now explicitly discuss the retrospective nature of the two studies in the overview of the current studies section of the manuscript (p. 5). 

P13 the participant details are too sparse to draw any conclusion about whether this was an unbiased and balanced sample. I’m not familiar with Murk but were the participants from a certain social, ethnic, demographic, background etc. Cloud research itself introduces bias.

Our original manuscript included standard demographic information on age, gender, and race. We now report all identifications for race in the revised manuscript (p. 6 and 9). Additionally, we include descriptive statistics for demographic information on political orientation, religiosity, social class background, and subjective socioeconomic status in a table in the supplementary materials on OSF. Across both studies, participants scored around the midpoint on each of these demographic variables on average.

The most important finding is that both studies demonstrated a return to “belief in free will” at pre pandemic levels. Ie freewill belief has not diminished.

I feel the middle period in both studies “the pandemic” was considered in too trivial a fashion. The authors talk about relinquishing in freewill because of overriding proven health reasons. As very little was and still is proven in this period this is a shaky assumption at best. Further they totally fail to address the different methods of coercion adopted by governments and bodies such as the WHO. The UK used subvertive advertising and a government nudge department. NHSE have had to subsequently admit that claims unvaccinated people spread covid more than vaccinated were untrue.

In both studies, participants were asked to write a brief description about their quarantine life (“during the beginning of the COVID-19 pandemic while under any state of emergency quarantine”). Most of the qualitative responses suggest that participants’ decline in belief in free will was due to a concerted effort to follow health guidelines to prevent the transmission of the COVID-19 virus. Common responses included working from home, attending school remotely, reducing social interactions, only leaving home for groceries, and monitoring the news more frequently as a means to evade the virus. Some participants indicated that they were already remote workers or not very social, so the pandemic did not change their day-to-day too much. Thus, it follows that this reduction in free will beliefs was likely due to a loss of personal agency from the unprecedented health crisis.

The reviewer brings up an interesting point about the implementation of COVID-19 pandemic guidelines. We recognize that there are differences in how governing bodies across the world responded to the COVID-19 pandemic. Since our samples are exclusively from the United States, we include this important point as a limitation and direction for future research in the Discussion (p. 13).

The authors were very honest and clear about the limitations of their study.

Thank you.

It would have been good if they could also have addressed that governments because of such findings of - no lasting damage to freewill - could use such unevidenced measures of lockdown again especially given their conclusion.

As our manuscript is focused on the resiliency of belief in free will using the COVID-19 pandemic as a unique context to test this research question, comments about the efficacy of government lockdowns may detract from the overall purpose of our research. We believe the last sentence in our concluding paragraph (“This sentiment is important to remember as we realize the consequences of this historical public health crisis in years to come.”) allows readers draw their own conclusions about how the COVID-19 pandemic has affected them and how to move forward if another public health crisis arises in the future.

The 2 figures illustrate their findings well and is powerful.

Thank you.

The paper is clear and simple. Perhaps over simplification is their enemy.

We thank the reviewer for this comment. We believe the revisions we made based on the helpful feedback from the reviewers has strengthened this manuscript.

---

## [Decision Letter · Decision Letter 1]

23 Apr 2024

At freedom’s edge: Belief in free will during the COVID-19 pandemic

PONE-D-23-31527R1

Dear Dr. Seto,

We’re pleased to inform you that your manuscript has been judged scientifically suitable for publication and will be formally accepted for publication once it meets all outstanding technical requirements.

Kind regards,

Ali B. Mahmoud, Ph.D.

Academic Editor

PLOS ONE

Additional Editor Comments (optional):

Reviewers' comments:

Reviewer's Responses to Questions

**Comments to the Author**

1. If the authors have adequately addressed your comments raised in a previous round of review and you feel that this manuscript is now acceptable for publication, you may indicate that here to bypass the “Comments to the Author” section, enter your conflict of interest statement in the “Confidential to Editor” section, and submit your "Accept" recommendation.

Reviewer #1: All comments have been addressed

2. Is the manuscript technically sound, and do the data support the conclusions?

Reviewer #1: Yes

3. Has the statistical analysis been performed appropriately and rigorously? 

Reviewer #1: Yes

4. Have the authors made all data underlying the findings in their manuscript fully available?

Reviewer #1: Yes

5. Is the manuscript presented in an intelligible fashion and written in standard English?

Reviewer #1: Yes

6. Review Comments to the Author

Reviewer #1: The ethics statement is repeated in the revised version. You could just add that the whole project was IRB approved.

All of my comments were adressed in this version.

7. PLOS authors have the option to publish the peer review history of their article (what does this mean?). If published, this will include your full peer review and any attached files.

Reviewer #1: **Yes: **Sidnei Rinaldo Priolo Filho

---

## [Editor Report · Acceptance letter]

14 May 2024

PONE-D-23-31527R1 

PLOS ONE

Dear Dr. Seto, 

I'm pleased to inform you that your manuscript has been deemed suitable for publication in PLOS ONE. Congratulations! Your manuscript is now being handed over to our production team.

Kind regards, 

on behalf of

Dr. Ali B. Mahmoud 

Academic Editor

PLOS ONE